# High Resolution Analysis of Proteome Dynamics during *Bacillus subtilis* Sporulation

**DOI:** 10.3390/ijms22179345

**Published:** 2021-08-28

**Authors:** Zhiwei Tu, Henk L. Dekker, Winfried Roseboom, Bhagyashree N. Swarge, Peter Setlow, Stanley Brul, Gertjan Kramer

**Affiliations:** 1Laboratory for Mass Spectrometry of Biomolecules, University of Amsterdam, Science Park 904, 1098 XH Amsterdam, The Netherlands; tuzhiwei2013@163.com (Z.T.); H.L.Dekker@uva.nl (H.L.D.); W.Roseboom@uva.nl (W.R.); B.N.Swarge@uva.nl (B.N.S.); 2Laboratory for Molecular Biology and Microbial Food Safety, University of Amsterdam, Science Park 904, 1098 XH Amsterdam, The Netherlands; 3Department of Molecular Biology and Biophysics, UCONN Health, Farmington, CT 06030-3305, USA; setlow@nso2.uchc.edu

**Keywords:** sporulation, proteomics, *Bacillus subtilis*

## Abstract

*Bacillus subtilis* vegetative cells switch to sporulation upon nutrient limitation. To investigate the proteome dynamics during sporulation, high-resolution time-lapse proteomics was performed in a cell population that was induced to sporulate synchronously. Here, we are the first to comprehensively investigate the changeover of sporulation regulatory proteins, coat proteins, and other proteins involved in sporulation and spore biogenesis. Protein co-expression analysis revealed four co-expressed modules (termed blue, brown, green, and yellow). Modules brown and green are upregulated during sporulation and contain proteins associated with sporulation. Module blue is negatively correlated with modules brown and green, containing ribosomal and metabolic proteins. Finally, module yellow shows co-expression with the three other modules. Notably, several proteins not belonging to any of the known transcription regulons were identified as co-expressed with modules brown and green, and might also play roles during sporulation. Finally, levels of some coat proteins, for example morphogenetic coat proteins, decreased late in sporulation.

## 1. Introduction

In response to unfavorable conditions, vegetative cells of *Bacillus subtilis* can enter the process of sporulation, to form resistant and metabolically dormant spores. A lot of effort has been made to investigate the mechanism of sporulation, and this process has been extensively reviewed in [1,2,3,4,5]. As a result of sporulation genes’ transcription and mRNA translation, the vegetative cells undergo a series of morphological changes until the release of spores into the environment. All the proteins necessary for a dormant spore are synthesized and deposited in the nascent spore during sporulation. The spore originates from the smaller compartment (forespore) of the asymmetrically divided cell formed in an early stage of sporulation. The forespore is then engulfed by the mother cell (the larger compartment). With the assembly of the cortex and coat layers surrounding the forespore, its core begins to dehydrate and takes up CaDPA (calcium dipicolinic acid) leading to further core dehydration. This process makes the forespores become bright under phase-contrast microscopy. Finally, the mature spore is released by lysis of the mother cell.

As expected, morphological changes in sporulating cells are highly correlated with gene and protein expression. Activation of specific sporulation transcriptional regulators takes place before forespore engulfment in the mother cell (SigE (RNA polymerase σ factor E)) and in the forespore compartment (SigF), and after forespore engulfment in the mother cell (SigK) and in the forespore (SigG) [6]. Several hours after completion of engulfment, metabolic activity in the forespore diminishes [7,8] and is largely, if not completely absent, in the free spore. However, studies from both Tu et al. [9] and Swarge et al. [10] have demonstrated the presence of many metabolic proteins in the free spores. Spores harboring varied levels of metabolic proteins exhibit different spore outgrowth properties [9]. RNA-seq has been a powerful tool in the analysis of sporulation by quantitation of levels of various mRNAs at various periods in this process [11,12]. However, a serious problem with the use of transcriptome or microarray data for research on sporulation is that the performers of spore assembly are proteins. Due to the fact that proteins and mRNA have different lifetimes their concentrations evolve on different time scales. As a consequence, inference of regulatory mechanisms from time series of mRNA data may potentially lead to incomplete conclusions.

To facilitate the investigation of protein changeover throughout the sporulation process, and thus directly interrogate the spore proteome, a *kinA*-inducible sporulation system has been set up [9,13]. This system allows more synchronous initiation of sporulation and leads to a significantly more homogeneous sporulating cell population. This in turn allows us to interrogate the sporulation process with a far greater sensitivity than is possible in the more heterogeneous populations created by traditional sporulation protocols. Using this newly established system we set out to characterize in great detail the sporulation process on the level of the proteome from the initiation of sporulation to the release of spores, in order to generate a highly time-resolved proteome map of *B. subtilis* sporulation. To this end we used metabolic labeling and mass spectrometry to monitor the proteome changeover during sporulation. This led to the quantification of a total of 2370 proteins. Co-expression network analysis revealed that there were four major modules of co-expressed proteins during sporulation. Two of these were mostly populated with sporulation-related proteins. In summary, this study shows for the first time a highly time-resolved view of protein expression changes during sporulation and reveals distinct modules of co-expressed proteins that are activated or repressed during the specific stages of sporulation.

## 2. Results

### 2.1. Morphological Stages during Sporulation

*KinA* was artificially expressed to induce sporulation in vegetative cells of strain 1887 of *Bacillus subtilis* PY79. During sporulation, phase-contrast and fluorescence microscopy images of cells with membranes visualized by the fluorescent membrane dye Nile Red were taken to monitor the morphological changes, and the proportion of different cell types was calculated (Figure 1). The asymmetric septum appeared immediately after the glucose dilution (time = 0 min) and was initially seen in approximately 10% of the population. The smaller forespore compartment would then be engulfed to become a phase-dark forespore. The proportion of cells exhibiting polar division, engulfment, and phase-dark forespores nearly reached a peak of 80% at 90 min. Phase-bright forespores appeared at 150 min (~in 30% of the cells) and reached ~80% of the population at 240 min. Free spores appeared at 300 min (0.36%) and reached a peak of 70% at 480 min. Vegetative cells that cannot be distinguished as any type of sporulating cell comprised at least 5% of the population at all time points.

### 2.2. Proteome Coverage of Sporulation

In total, 23 samples from 0 to 480 min were harvested from each of three independent replicates to perform proteomic analysis. Overall, 2370 proteins were quantified in the analysis (Figure 2A), and 428 proteins were quantified in all the timepoints with at least one quantification at every timepoint. The proteins were assigned to their transcriptional regulons according to SubtiWiki [14] (Figure 2B). Among them, key regulons (Spo0A, SigE, SigF, SigG, and SigK) of sporulation make up a high proportion of the proteins.

### 2.3. Sporulation Regulatory Proteins

Sporulation is controlled by a hierarchical regulatory network [6]. It involves sporulation master regulator Spo0A initiating sporulation and directly and indirectly controlling forespore and mother cell specific regulators, SigE, SigF, SigG, SigK. All these 5 proteins were quantified in some of the timepoints but failed to show clear expression profiles (Appendix A, see Appendix A). In the phosphorelay involved in phosphorylating Spo0A, KinA was quantified in all timepoints, and its protein amount gradually decreased (Figure 3). Other phosphorelay proteins were also quantified in a number of timepoints (Appendix A). Mother cell early sporulation regulators GerR and SpoIIID are transcriptionally activated by SigE and positively or negatively regulate sporulation. Both proteins were quantified to upregulate in the first 60 min and gradually decreased after 180 min (Figure 3). SpoIIIAH and SpoIIQ are components of a feeding tube connecting mother cell and forespore and are important for SigG activation [6,15]. They were upregulated shortly after sporulation initiation, reached a peak amount at 90–105 min, and then decreased (Figure 3). The SigK-activated mother cell late sporulation regulator GerE was upregulated after 120 min and the SigG-activated forespore regulator SpoVT was upregulated after 90 min (Figure 3).

### 2.4. Spore Coat Proteins, Small Acid-Soluble Spore Proteins (SASPs), and Other Sporulation Proteins

The spore coat proteins start to express and assemble upon completion of asymmetric division [2,5]. A precoat structure is first assembled to serve as a scaffold for subsequent coat assembly [16,17]. The scaffold includes at least several morphogenetic proteins (SpoIVA, SpoVID, SpoVM, SafA, and CotE) which are crucial for subsequent coat layer assembly. SpoIVA, SpoVID, and SpoVM are the first upregulated morphogenetic proteins (Figure 4A), and mutation of any of these three proteins causes improper assembly of overall coat layers [18,19,20]. SafA, the morphogenetic protein for the inner coat [21], is upregulated after the expression of SpoIVA, SpoVID, and SpoVM (Figure 4A). CotE, the morphogenetic protein for the outer coat [22], is expressed following SafA (Figure 4A). CotX, CotY, and CotZ are morphogenetic proteins for the outmost crust layer [23,24], and are the last expressed morphogenetic proteins (Figure 4A). CotO and CotH also play roles in the proper formation of the outer coat and cooperate with CotE [16,25,26], but the expression of CotO and CotH is earlier than CotE (Figure 4B). In terms of coat protein assembly, six classes of coat proteins have been identified according to their localization dynamics [14,27]. Quantified proteins of the six classes are visualized in Figure 4C. Four distinct time ranges are recognized for the upregulation of coat proteins, 0–45 min, Class I; 45–105 min, Class II; 105–180 min, Classes III, IV, and V; after 180 min, Class VI; see Appendix A for more information on this topic. Based on the determined time ranges, coat proteins with unassigned classes can be classified into different clusters of candidates for the six classes (Figure 4D). A remarkable feature of the coat protein expression is the decrease of the amounts of some coat proteins in the later stage of sporulation, for example proteins in Figure 4A,C. In the quantification of SASPs, 10 of 16 SASPs showed clear expression profiles (Figure 4E). SspG increased after 210 min and others increased at 60–105 min.

Besides sporulation regulatory proteins, spore coat proteins, and SASPs, there are a number of other sporulation proteins with known and unknown functions. These proteins are involved in metabolism, resistance to some stresses, and some other sporulation processes. These sporulation proteins were manually classified into seven clusters with different expression patterns (Figure 5). Some of the proteins are only upregulated in intermediate stages of sporulation and some gradually decrease during sporulation.

### 2.5. Protein Co-Expression Analysis

Proteins showing similar expression profiles are considered to be controlled by the same regulatory system or play roles in related functions. To investigate overall protein expression and identify possible unreported sporulation proteins, a protein co-expression analysis was performed using WGCNA [28]. Proteins showing similar expression profiles are clustered into the same co-expressed modules. Four co-expressed modules were detected in the protein co-expression network analysis, where modules are annotated by different colors (Figure 6A). The number of proteins per module were 267 (blue), 93 (brown), 27 (green), and 42 (yellow). Appendix A lists the protein members of modules. Figure 6B shows the co-expression network of the most abundant co-expressed proteins. Proteins in modules brown and green are upregulated during sporulation. They are involved in sporulation and are members of the SigG, SigK, SigE, SigF, GerE, and GerR regulons (Appendix A). Proteins in module blue are downregulated during sporulation, and contain proteins involved in metabolism and translation and are mainly regulated by the stringent response (Appendix A). Module yellow proteins are involved in metabolism and some of these proteins are regulated by the stringent response and SigB (Appendix A). Proteins not included in regulons but co-expressed within sporulation modules brown and green are identified and listed in Table 1. Perhaps these proteins also have important, but as yet unknown roles in sporulation.

## 3. Discussion

Spores are assembled in a process named sporulation, and spores’ properties and protein composition are highly correlated with sporulation conditions [30,31,32]. To better understand the influence of the sporulation process on spore properties, a time series proteomic analysis is essential. However, sporulation initiation is highly heterogenous. The use of a *kinA*-inducible strain of *B. subtilis* makes the sporulating cell population more homogenous and initiation of sporulation controllable [9]. Samples were harvested immediately following the culture dilution which was 90 min after *kinA* was induced. Therefore, the proteome changes during the time of *kinA* induction to Spo0A phosphorylation and thus the threshold of sporulation needs additional study. Transcription of *kinA* is induced by a constant concentration of IPTG (Isopropyl-D-1-thiogalactopyranoside). However, the amount of KinA gradually decreased during sporulation. This could be due to the downregulation of proteins involved in translation in module blue.

From microscopy results, we can clearly see that sporulation initiated quickly at timepoint 0, as free spores appeared at 300 min and reached 70% at 480 min. During this sporulation process, the overall proteome changeover was investigated in depth for the first time. We validated expression of sporulation regulatory proteins, coat proteins, and SASPs, as well as other proteins involved in sporulation. In addition, through protein co-expression analysis, a number of proteins co-expressed within modules of sporulation were identified and could be considered as new proteins potentially involved in sporulation. Besides the sporulation modules (brown and green), module blue comprised quite a number of proteins involved in metabolism, translation, synthesis of antibiotics, and amino acids. It shows an opposite expression pattern (gradually decreasing in protein amounts) compared to proteins in modules brown and green. However, they are connected through module yellow in the co-expression network analysis, indicating that regulatory systems of module blue and sporulation modules brown and green may have some sort of cooperation to modulate other activities in sporulation, such as metabolism and protein translation. Remarkably, amounts of some coat proteins, for example morphogenetic coat proteins, decreased late in sporulation. At present it is unclear why the levels of these coat proteins decreased, and no previous studies have reported this phenomenon. A logical assumption is that such coat proteins could have played roles in guiding or helping assembly of other coat proteins, after which they became “surplus” and were degraded.

An outstanding question about sporulation is how spore coat components are dynamically expressed. In a transcriptomics profiling study of *Bacillus anthracis*, it was reported that the genes whose products make up the spore proteome were overrepresented in an earlier phase of the life cycle, indicating that the majority of spore proteins are packaged from preexisting stocks rather than synthesized de novo [33]. In our data, all the coat proteins and SASPs were upregulated after sporulation initiation. Only a small number of sporulation proteins showed a decrease during sporulation, indicating they may be present in excess prior to the beginning of sporulation. However, a minimal set of metabolic proteins are enclosed in mature spores [9,10]. Most of these metabolic proteins are members of module blue having a decreased expression profile during sporulation. In coat assembly, some of the coat proteins may also be expressed prior to their assembly, for example, CotO and CotH are associated with CotE in assembly, but are expressed earlier than CotE.

All in all, this study shed light on the dynamics of protein expression during sporulation at high temporal resolution and shows it to be a highly dynamic process. A protein co-expression analysis revealed the global organization of protein expression during sporulation.

## 4. Materials and Methods

### 4.1. Sporulation and Sampling

The *KinA*-inducible *B. subtilis* strain 1887 was used in this study, and *kinA* transcription relies on the presence of IPTG (Sigma-Aldrich, St. Louis, MI, USA) in the culture medium [13]. The sporulation protocol described in [9] was applied in this study. In short, vegetative cells were grown at 37 °C under continuous agitation (200 rpm) in a MOPS (3-[N-Morpholino]propanesulfonic acid, Sigma-Aldrich, St. Louis, MI, USA)-buffered medium containing 40 mM of NH_4_Cl and 40 mM of glucose. At exponential phase (OD_600_ = 0.65), the cells were induced to express *kinA* with 100 µm of IPTG for 90 min. Subsequently, the culture was diluted six-fold with the same, pre-warmed medium now devoid of glucose. Time 0 min was defined as the moment after culture dilution. For proteomic analysis, 10 mL of culture was sampled at 0, 15, 30, 45, 60, 75, 90,105, 120, 135, 150, 165, 180, 210, 240, 270, 300, 330, 360, 390, 420, 450, and 480 min. Three biological replicates were harvested at each time point. Harvested samples were stored at −80 °C before further experiments. For microscopy, 500 µL of culture was taken at 0, 30, 60, 90, 120, 150, 180, 240, 300, 360, 420, and 480 min from replicate 1 to monitor the process of sporulation.

### 4.2. Microscopy

Harvested cells were concentrated by centrifugation at 10,000 rpm for 1 min and immediately loaded on a 3% *w*/*v* agarose (in water) pad on a microscope slide supplemented with 5 µg/mL of membrane dye Nile Red (Invitrogen, Paisley, UK). The agarose pad was made following the protocol described previously [34]. Phase-contrast and fluorescent images were taken with a Nikon Eclipse Ti microscope. The membrane dye was visualized at an excitation wavelength of 570 ± 10 nm and emission wavelength of 620 ± 10 nm. The Nikon Eclipse Ti was equipped with an Intensilight HG 130 W lamp, A Nikon CFI Plan Apo Lambda 100× oil objective, C11440-22CU Hamamatsu ORCA flash 4.0 camera, LAMBDA 10-B Smart Shutter form Sutter Instrument, an OkoLab stage incubator, and NIS elements software version 4.50.00. Microscopy images were analyzed with ImageJ/Fiji (http://fiji.sc/Fiji (accessed on 6 February 2020)) [35].

### 4.3. LC-MS/MS

A ^15^N metabolic labeling strategy was used to perform proteomic analysis. The strategy applied was previously described by Abhyankar et al. [36]. Briefly, an aliquot of 0.5 mL at an OD_600_ of 1 ^15^N-labeled vegetative cells at exponential phase (OD600 = 0.65) and 1.5 mL at on an OD_600_ of 1 ^15^N-labeled free spores was mixed with every harvested sample. The vegetative cells and spores were made using *B. subtilis* strain 1887. Induced spores were harvested at 8 h after the dilution and were purified from the remaining vegetative cells with Histodenz (Sigma–Aldrich, St. Louis, MO, USA) density gradient centrifugation [37]. The mixed samples were processed including protease digestion following the “one-pot” sample processing method [38].

Tryptic samples were reconstituted in 0.1% formic acid in water and 200 ng of equivalent (determined by measuring the absorbance at a wavelength of 205 nm [39]) was injected by a Ultimate 3000 RSLCnano UHPLC system (Thermo Scientific, Germeringen, Germany) onto a 75 µm × 250 mm analytical column (C18, 1.6 µum particle size, Aurora, Ionopticks, Melbourne, Australia) kept at 50 °C at 400 nl/min for 15 min in 3% solvent B before being separated by a multi-step gradient (Solvent A: 0.1% formic acid in water, Solvent B: 0.1% formic acid in acetonitrile) to 5% B at 16 min, 17% B at 38 min, 25% B at 43 min, 34% B at 46 min, and 99% B at 47 min held until 54 min returning to initial conditions at 55 min equilibrating until 80 min.

Eluting peptides were sprayed by the emitter coupled to the column into a captive spray source (Bruker, Bremen Germany) with a capillary voltage of 1.5 kV, a source gas flow of 3 L/min of pure nitrogen and a dry temperature setting of 180 °C, attached to a timsTOF pro (Bruker, Bremen Germany) trapped ion mobility, quadrupole, time-of-flight mass spectrometer. The timsTOF was operated in PASEF (parallel accumulation–serial fragmentation) mode of acquisition. The TOF scan range was 100–1700 m/z and a tims range of 0.6–1.6 V.s/cm^2^. In PASEF mode, a filter was applied to the m/z and ion mobility plane to select features most likely representing peptide precursors, the quad isolation width was 2 Th at 700 m/z and 3 Th at 800 m/z, and the collision energy was ramped from 20–59 eV over the tims scan range to generate fragmentation spectra. A total of 10 PASEF MS/MS scans scheduled with a total cycle time of 1.16 s, scheduling target intensity 2 × 10^4^ and intensity threshold of 2.5 × 10^3^ and a charge state range of 0–5 were used. Active exclusion was on (release after 0.4 min), reconsidering precursors if ratio current/previous intensity >4.

### 4.4. Data Processing and Bioinformatics

The raw data were processed with MASCOT DISTILLER (version 2.7.1.0, 64 bits), MDRO 2.7.1.0 (MATRIX science, London, UK), including the Search toolbox and the Quantification toolbox. Prior to processing, data were converted from Bruker’s tdf format to Bruker’s baf format by a script provided by Bruker to make raw files compatible with MASCOT DISTILLER. Peaks were fitted to a simulated isotope distribution, with a correlation threshold of 0.6, with minimum signal to noise ratio of 1.3. The processed data were searched in a MudPIT approach with MASCOT server 2.7.0 (MATRIX science, London, UK), against the *B. subtilis* 168 ORF translation database. The MASCOT search parameters were as follows: enzyme—trypsin, allowance of two missed cleavages, fixed modification—carbamidomethylation of cysteine, variable modifications—oxidation of methionine and deamidation of asparagine and glutamine, quantification method—metabolic ^15^N labeling, peptide mass tolerance and peptide fragment mass tolerance—50 ppm. Using the quantification toolbox, the quantification of the light peptides relative to the corresponding heavy peptides was determined as an ^14^N/^15^N ratio, using Simpson’s integration of the peptide MS chromatographic profiles for all detected charge states. The quantification parameters were: correlation threshold for isotopic distribution fit—0.7, ^15^N label content—99.6%, XIC threshold—0.1, all charge states on, max XIC width—250 s, elution time shift for heavy and light peptides—20 s. The protein isotopic ratios were then calculated as the median over the corresponding peptide ratios.

The output quantitative data with a protein MASCOT score higher than 20 were considered as reliable. Proteins quantified in at least 46 of 69 samples were selected for the protein co-expression analysis. The missing values in the selected proteins were imputed using R software package Amelia [40]. Log_2_ transformation was applied to the data prior to the imputation. Average values over three biological replicates were used for co-expression network analysis utilizing R/Bioconductor software package WGCNA [28]. R scripts can be found in Appendix A. The signed network was used in the analysis. The soft threshold and the minimum module size in the analysis were 9 and 15, respectively. Transcriptional regulators of genes, their products, and functions were determined according to SubtiWiki (http://subtiwiki.uni-goettingen.de/ (accessed on 16 December 2020)) [14]. Lists of coat proteins and other proteins involved in sporulation were also acquired from SubtiWiki.

## Figures and Tables

**Figure 1 ijms-22-09345-f001:**
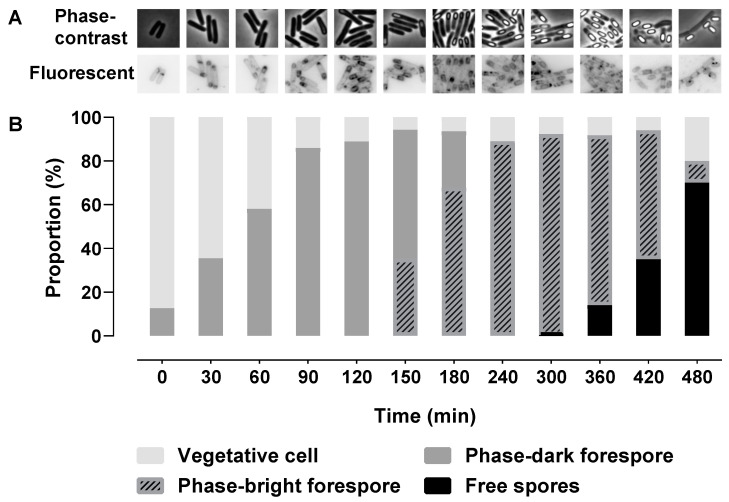
Morphological images of the sporulation process (**A**) and the proportions of different cell types (**B**). Time 0 is the moment after the cell culture is diluted. Phase-dark forespores include cells exhibiting polar division, engulfment, and engulfed forespores before becoming phase-bright. Fluorescent images were visualized with the membrane dye Nile Red.

**Figure 2 ijms-22-09345-f002:**
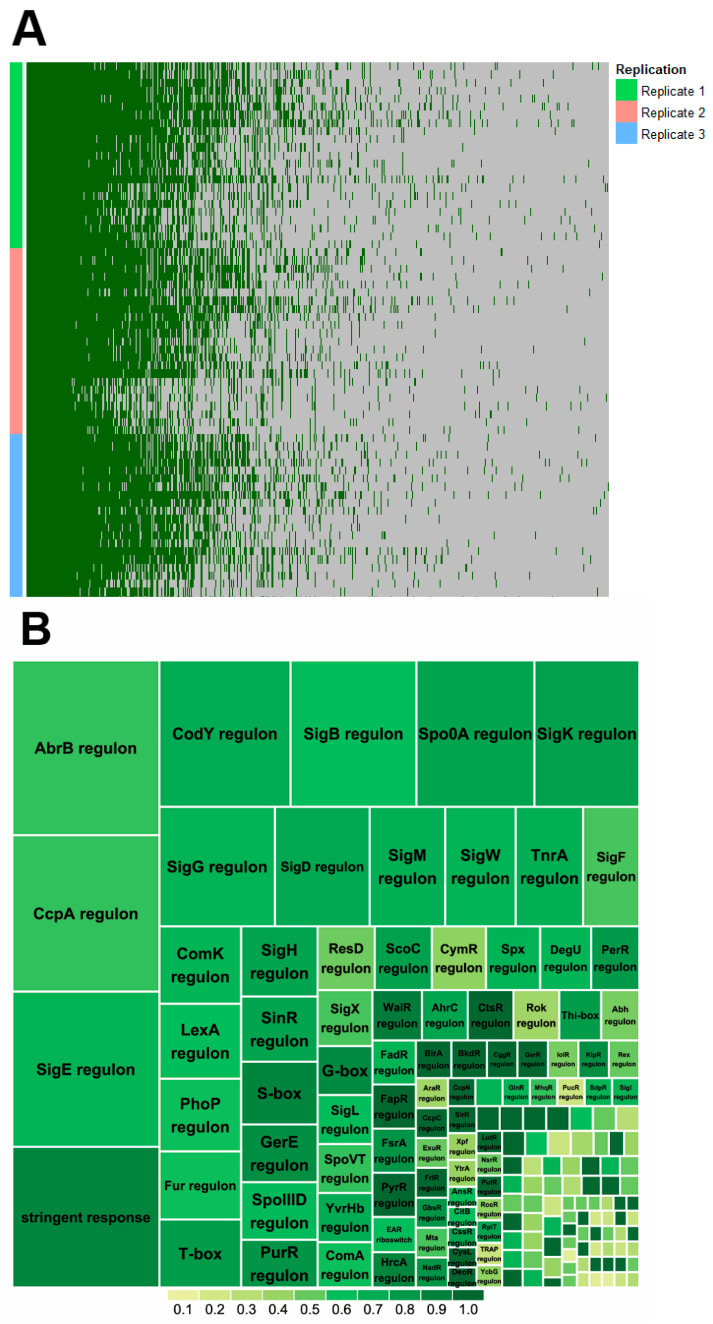
Overview of the quantification of proteins during sporulation (**A**) and their involved transcriptional regulons (**B**). (**A**) Visualization of data completeness of the quantified proteins. Columns indicate proteins, in total 2370 proteins, and rows indicate samples from three replicates with 23 timepoints in each (from top to bottom in each replicate are timepoints from 0 to 480 min). Dark green and grey indicate observed and missing values, respectively. (**B**) Tree-map summary of regulons of the quantified proteins. Sizes of the various rectangles correspond to the number of quantified proteins controlled by the regulon; the color of the squares indicates the ratio of the fractions of the quantified proteins belonging to the regulon to the total protein number in the regulon according to the color legend. Regulon names are not shown throughout for graphical reasons. Note that proteins could occupy more than one regulon.

**Figure 3 ijms-22-09345-f003:**
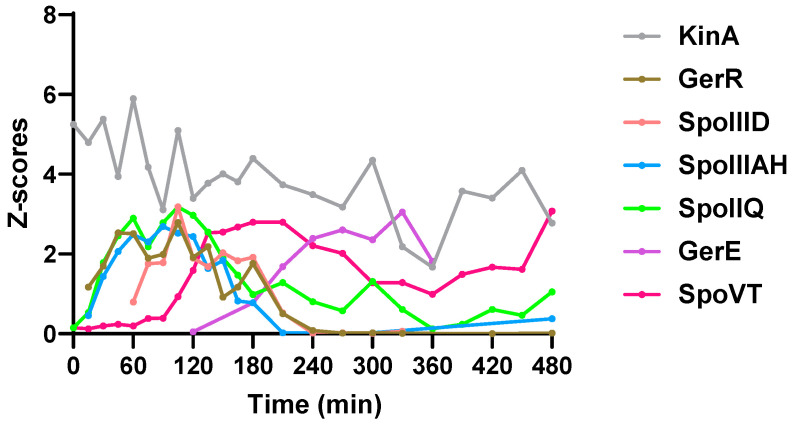
Changes in levels of sporulation regulatory proteins. KinA is artificially induced to express and finally phosphorylates Spo0A through Spo0F and Spo0B. GerR and SpoIIID are two SigE controlled mother cell regulators. SpoIIQ and SpoIIIAH are components of a feeding tube between mother cell and forespore. GerE is a SigK-controlled mother cell regulator. SpoVT is a SigG-controlled forespore regulator. Z-scores on the Y-axis represent the quantified value of every protein normalized to have a standard deviation of 1.

**Figure 4 ijms-22-09345-f004:**
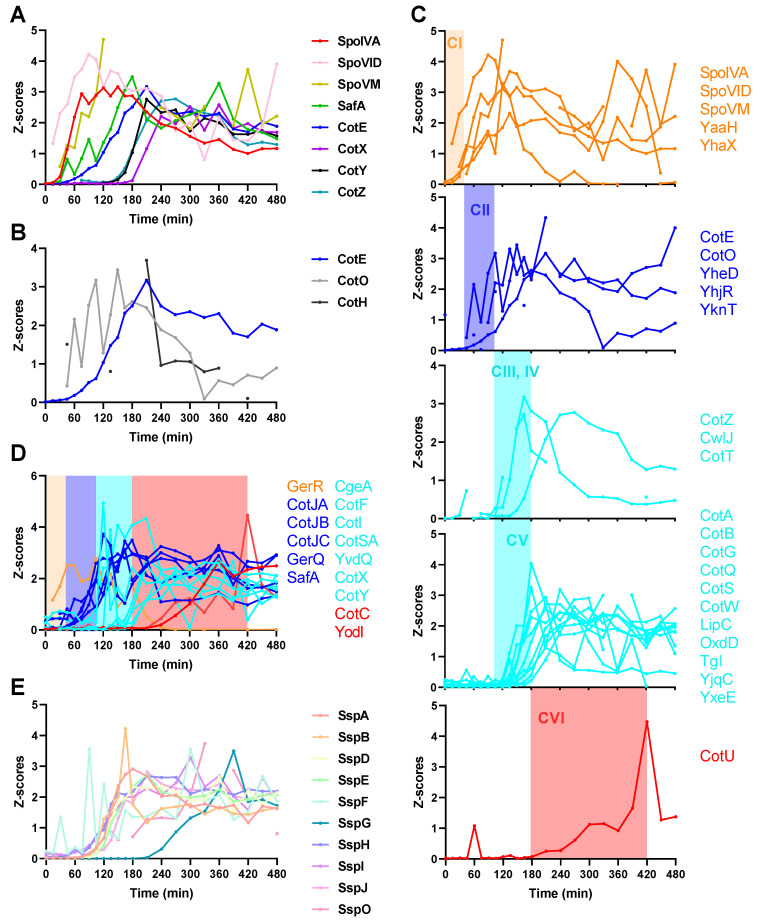
Expression of spore coat proteins and small acid soluble proteins. For point-to-point connection, a gap is left at each missing value. (**A**) Expression of morphogenetic coat proteins. (**B**) Expression of CotE, CotO, and CotH. CotO is expressed earlier than CotE. CotH has a higher level of protein than CotE at 45 min. (**C**) Classification of coat proteins according to their localization dynamics. Adapted from SubtiWiki (http://subtiwiki.uni-goettingen.de/ (accessed on 16 December 2020).) [14]. Class I (CI), early localizing spore coat proteins cover the outer forespore membrane; Class II (CII), early localizing spore coat proteins begin to encase the spore only after engulfment is complete; Classes III, IV (CIII, IV), early and late localizing spore coat proteins start to encase the spore only after completion of engulfment and the appearance of phase dark spores; Class V (CV), late localizing spore coat proteins localize exclusively to phase dark spores; Class VI (CVI), late localizing spore coat proteins localize exclusively to phase bright spores. Proteins upregulated in a similar time period are denoted with the same color. (**D**) Classification of the unassigned coat proteins. (**E**) Expression of small acid-soluble spore proteins (SASPs). Z-scores on the Y-axis represent the quantified value of every protein normalized to have a standard deviation of 1. Note that detailed information on the expression levels of all these individual proteins can be found in Appendix A.

**Figure 5 ijms-22-09345-f005:**
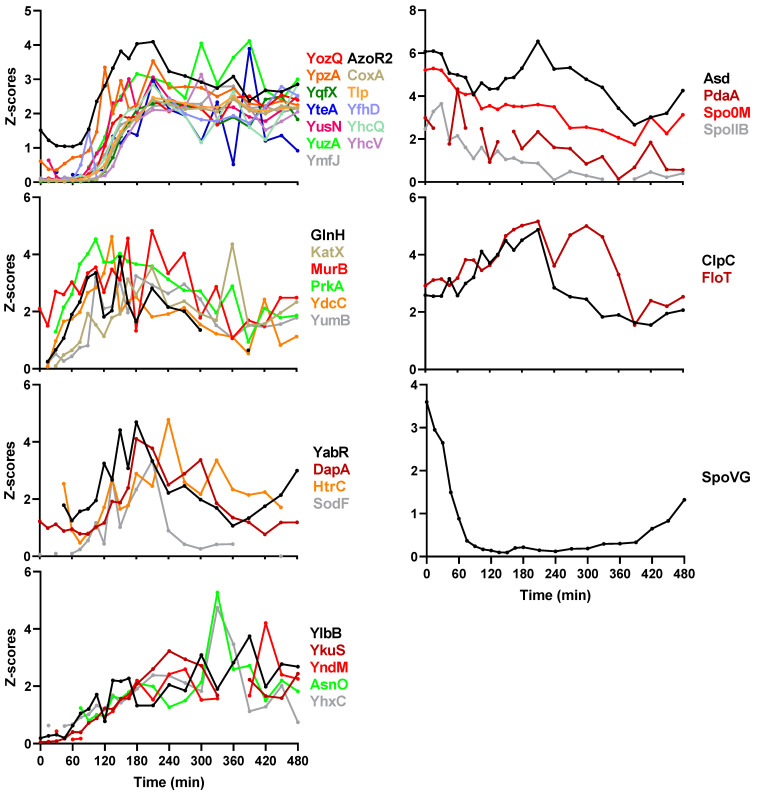
Classification of other sporulation proteins. Z-scores on the Y-axis represent the quantified value of every protein normalized to have a standard deviation of 1.

**Figure 6 ijms-22-09345-f006:**
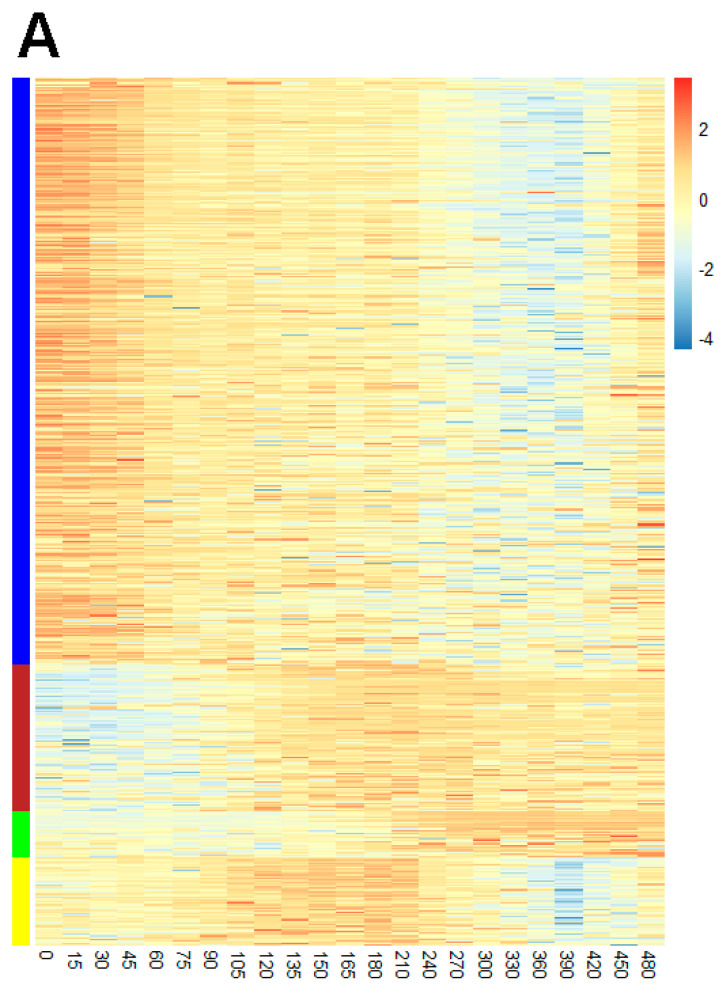
Heatmap visualization of the protein expression profiles (**A**) and network of the most abundant co-expressed proteins (**B**). The color in (**A**) represents Z-transformed expression values according to the color legend. Columns and rows indicate timepoints (specified on the bottom) and proteins, respectively. The assigned modules are colored on the left side. (**B**), generated by Cytoscape (version 3.8.0) [29]. Nodes represent expression profiles of corresponding proteins and edges indicate the proteins, which to some extent, are co-expressed. The edges shown have a threshold of greater than 0.2.

**Table 1 ijms-22-09345-t001:** Co-expressed proteins in modules brown and green without assigned regulons.

Module	UniProt ID	Protein	Function	Product
Brown	P50727	Fer	Electron transfer	Ferredoxin
Brown	O31796	Hfq	Unknown	RNA chaperone
Brown	O07609	YhfK	Unknown	Unknown
Brown	P45872	PrfA	Translation	Peptide chain release factor 1
Brown	P94521	YsdC	Unknown	Unknown
Brown	P54550	YqjM	Reduction of double bonds of nonsaturated aldehydes	NADPH-dependent flavin oxidoreductase
Brown	O34389	MalS	Malate utilization	Malate dehydrogenase (decarboxylating)
Brown	P28619	Rph	3–5 exoribonuclease	RNase PH (EC 2.7.7.56)
Brown	O34503	YtzD	Unknown	Unknown
Brown	P94425	YcnE	Unknown	Unknown
Brown	O31509	YeeI	Unknown	Unknown
Brown	C0H447	YpzJ	Unknown	Unknown
Brown	O34600	NrnA	Degradation of RNA oligonucleotides	Oligoribonuclease (nanoRNase), 3,5-bisphosphate nucleotidase
Brown	P46343	PhoH	Unknown	Unknown
Brown	P70949	YitW	Assembly of iron-sulfur clusters	Iron-sulfur cluster assembly factor
Green	P49778	Efp	Translation	Elongation factor P
Green	Q45495	Def	Utilization of S-methyl-cysteine	N-formylcysteine deformylase
Green	P54457	YqeL	Translation	Ribosomal silencing factor
Green	P40737	YxxD	Inhibition of the cytotoxic activity of YxiD	Antitoxin
Green	O31976	YomI	Cell wall turnover	Cell wall hydrolase

## Data Availability

The raw LC-MS data and output files can be found at the Massive Repository for Mass Spectrometry data (MSV000087154) or at ProteomeXchange (PXD025157).

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
