# Peer review of "High Resolution Analysis of Proteome Dynamics during Bacillus subtilis Sporulation"

_ijms, 2021, doi:10.3390/ijms22179345_

Round 1

Reviewer 1 Report

1.  I suggest few formal changes, e.g.

- Line 20  “…. four co-expressed modules (blue, brown, green and yellow).” to “…four co-expressed modules (termed blue, brown, green and yellow).”

 - Line 52 “Both [9] and a study from Swarge et al. [10]” to “Both study from Tu et al. [9] and Swarge et al. [10]”

- Line 245 “At present it is unclear why these coat proteins needed to decrease” – I suggest change of this formulation, which uses personification of proteins

2. Fig.1A The quality of fluorescent images can be improved. There are some “shades” in the images, probably from contrast enhancements or caused by the import of figures.

3. Fig. 2B The visibility of some regulons is bad, please increase the size of the image and there is a mistake “tringent response”

4. Fig. 6B I suggest to enlarge also this figure to make it readable, and change font colours (e.g. to white font in blue boxes…)

5. Line 100 “In total, 23 samples from 0 to 480 min were harvested from each of the three replicates to perform proteomic analysis.”

- Were the 3 replicates performed as 3 independent experiments? If not, additional independent repetition of the analysis (e.g. again with 3 replicates) could fill some missing time-points and validate some ambiguous data. For instance, in the case of CotO the up and down expression profile during first 180 min. It can be interesting to find out if its expression is cyclic or the fluctuations originate from measurement.

6. Line 101 “428 proteins were quantified in all the timepoints with at least one quantification at every timepoint.”

- Is it true for all expression profiles showed in the manuscript? For instance, SpoVM expression profile is missing between time-points 120-240 and  300-390 min and also in case of several other proteins in Fig. 4C there are missing time-points. Were they excluded from the study for some reason?

7. I understand that Fig. 4C intends to be illustrative about protein groups with similar expression trends but it could bring additional detailed information for the readers if different legends for individual proteins are used, e.g. at least different shades or line dashings.

8. How many of the identified proteins were transmembrane proteins? It is possible that some membrane proteins could not be identified by the used method but can be of great interest for the readers. Did you try to increase the proportion of the membrane proteins in the protein samples, e.g. by some enrichment methods as subcellular fractionation method?

9. Line 244 “Remarkably, amounts of some coat proteins, for example morphogenetic coat proteins, decreased late in sporulation. “

- It can be interesting for readers to get more detailed information about particular proteins, which show decreased amounts and if the obtained data are in agreement with transcription profiles known from the literature. For instance, SpoIVA and SafA have not only decreased expression but also transcription levels. So protein degradation might not be involved.

10. Is the increased SpoVID amount in the last time point relevant/significant? How authors explain this increase?

Reviewer 2 Report

General comment

Tu and co-workers induce synchronous sporulation to investigate the proteome dynamics during spore formation. The authors document the changeover of sporulation regulatory proteins, coat proteins, and other proteins involved in sporulation and spore biogenesis. It is a very nice job, although the title of the article was more provocative “High-resolution analysis of proteome dynamics during Bacillus subtilis sporulation” than the information information. I was expecting more from the manuscript, as a relative quantification of the number of specific proteins per cell, and the address of another relevant question, that is the proteomic changes during the resuscitation of the formed spore. Can a mature spore hold all those proteins listed in Table 1? Why relative abundant proteins are not included among the 2371 proteins listed? Why does the initiation of replication start earlier on a reviving spore if the replication proteins are packed in the forming spore? How can a resuscitating spore repair a DNA double-strand break through a ligation-based pathway (non-homologous end-joining) if end processing proteins (AddAB) are packed in the spore to levels comparable to 90 min after induction of sporulation (time 0)?

Specific comments,

  1. Lanes (L) 25-28. Are necessary the sentences that initiate with “we speculate”? I suggest deleting them or rephrase.
  2. L 54-61, the authors open expectations that are not fully filled. RNA-seq quantified the mRNA level at various periods, but as well pointed, proteins and RNAs have different lifetimes. However, this statement is not further pursued. For example, the fate of those proteins with the longer-living mRNA is poorly explored (Hederstedt and co-workers 2003).
  3. L 101. It is stated, “Overall, 2370 proteins were quantified in the analysis . . . 428 proteins were quantified in all the timepoints . . . “. In figure 2A the analysis was qualitative, and in Table 1 it is hard to interpret the presented data. First, in Table 1, for protein 1 at time 60 min (sp|O31423|SfkB) there is a number of 29,97 (or should be 29.97), whereas in protein 183 at time zero (sp|Q7WY59|SspG) there is a number of 0,000947. What does this mean? Does the assay have a 3.2 x 10E4-fold sensitivity? How many SspG molecules per cell are at time zero? Is reliable a 6 digit interval for such small number of SspG? Second, protein identification is complicated. In Table 1, protein 2 is stated “sp|P02968|FLA_BACSU”, which is encoded by which gene? BACSU repeated in every protein could be stated once in the Table legend and the gene name should be easily visible. Third, the few proteins that I have tested, are listed in Table 2 but not in Table 1. Fourth, why the 430 proteins listed in Table 2 were quantified with numbers below 3. Fifth, protein 141 (Table 2, sp|P37455.SsbA_BACSU) and protein 451 (Table 1, sp|P16971|RecA) at time 0 min should be abundant (at least during late exponential phase) but are presented with an average of 1.71 and 3.76, respectively. In contrast, proteins 95 and 99 (Table 1, sp|P37871|RpoC and sp|P37870|RpoB) which have an average value of 2.43 and 2.38 should be significantly less abundant than SsbA at time 0 min. Is my interpretation correct?
  4. L 101-102. It is stated “. . . key regulons (Spo0A, SigE, SigF, SigG and SigK) of sporulation make up a high proportion of the proteins”. I was searching for their quantitative values in Table 1 at time zero, and I have found for AbrB and Spo0A (identification numbers of P08874 and P06534) quantification numbers of 0.88 and 4.31. Can I deduce that at time zero there is about 5-fold more Spo0A than AbrB? Then, I search for the quantitative information in Figure 2B, and other regulators have bigger quadrilateral sizes than these ones. Please improve the presentation of the data, and explain the discrepancies. Why a rectangle is termed square?
  5. Figure 3. I have trouble understanding the levels of expression of the depicted proteins. In Table 1 at time zero KinA has a quantification number of 12.82 and SpoIIQ of -1.16, but in Figure 3 the difference between these proteins is smaller than factor 100.
  6. Figure 4. Is SpIVD in Figure 4A SpoIVD? Are SpIVA, SpVID and SpVM in Figure 4N SpoIVA, SpoVID and SpoVM? In Figure 4A, the colors used for SpoIVA, SpoVID, and CodY are not easy to differentiate.
  7. Figure 5. A black and white figure with many, many entries is hard to differentiate which is which.
  8. Figure 6. The resolution of Fig. 6B is too load to be able to read the proteins involved.
  9. L 201-202. Can a reader infer that the number of RNA polymerase molecules is nearly constant in the 480 min assay? Is this implying that spores have an excess of RNA polymerase molecules? I was missing information about cell physiology. For example, if there is a DNA double-strand break during the late stage of sporulation, it can be repaired through ligation-based pathways (non-homologous end-joining) rather than by replication-based pathways (but the AddAB protein levels remain constant between 0 and 480 min).
  10. L 238 and Table 1. The levels of RpoB and RpoC proteins strongly decreased (between 270 – 450 min and 270 - 420 min, respectively), but increased at 480 min, but the levels are below those observed at time 0 min (1.6-fold and 4-fold, respectively). For spore resuscitation, the RNA polymerase should be packed in the mature spore.

Round 2

Reviewer 1 Report

The authors replied to all of my questions and made changes in the manuscript that improved the visibility and resolution of the presented data. I recommend to accept the manuscript for the publication.

Author Response

We are happy the reviewer now approved the manuscript.

Reviewer 2 Report

to the query of L 201 -202 the authors respond that "The "AddA and AddB" proteins likely are present in spores for later use during germination and genome duplication preceding first cell division. I though that DNA double-strand breaks are repaired through ligation-based pathways during the germination stage (by Ku and LigD) and the decision that govern this outcome is should be in competition with AddAB-mediated end resection. The pathway choice hypothesis does not fit with the presented results.

Author Response

Of the proteins typically involved in nonhomologous end joining, SbcC (O06714) and SbcD (P23479) do not have enough values to show a clear expression trend. PNPase (P50849) shows some variation, but nearly remains constant. Hence we conclude that these proteins are expressed and might be functional in double strand break repair at late stages of sporulation although there are no indications for their enhanced presence at these time points.